# Novel Modification of a Confirmatory SMA Sequencing Assay that Can Be Used to Determine *SMN2* Copy Number

**DOI:** 10.3390/ijns7030047

**Published:** 2021-07-21

**Authors:** Binod Kumar, Samantha Barton, Jolanta Kordowska, Roger B. Eaton, Anne M. Counihan, Jaime E. Hale, Anne Marie Comeau

**Affiliations:** 1New England Newborn Screening Program, University of Massachusetts Medical School, Worcester, MA 01605, USA; Samantha.Barton@umassmed.edu (S.B.); Jolanta.Kordowska@umassmed.edu (J.K.); roger.eaton@umassmed.edu (R.B.E.); anne.counihan@umassmed.edu (A.M.C.); jaime.hale@umassmed.edu (J.E.H.); anne.comeau@umassmed.edu (A.M.C.); 2Division of Genetics, Department of Pediatrics, University of Massachusetts Medical School, Worcester, MA 01605, USA

**Keywords:** newborn screening, SMA, sequencing, copy number, *SMN2*

## Abstract

Promising treatments for spinal muscular atrophy (SMA), the leading genetic cause of infant mortality, prompted calls for inclusion in newborn screening (NBS). In January 2018, the New England Newborn Screening Program (NENSP) began statewide screening for SMA using a tiered algorithm looking for the absence of *SMN1* Exon 7. When results from the first and second tier needed reconciliation, we developed and validated a third tier DNA sequencing assay to ensure the presence or absence of *SMN1* Exon 7. All nine infants referred to specialty centers through NBS showed single base substitution of c.840C>T, and were confirmed to have SMA. Further, a minor sequencing protocol modification allowed the estimation of *SMN2* copy number in SMA affected patients; we developed and validated a copy-number assay yielding 100% match with seven previously characterized specimens of SMA patients. All nine SMA-affected infants found through NBS were also assayed for *SMN2* copy number. Results were comparable but not 100% matched with those that were reported by independent diagnostic laboratories. In conclusion, a sequencing protocol confirms NBS findings from real-time qPCR, and its modified application allows NBS programs that have sequencing capabilities to provide *SMN2* copy numbers without the need for additional instrumentation.

## 1. Introduction

Spinal muscular atrophy (SMA) is an autosomal recessive disorder characterized by degeneration of the motor neurons of the spinal cord, which causes proximal paralysis with muscular atrophy. SMA is associated with the absence of a functional gene called Survival of Motor Neuron 1, *SMN1* (telomeric), and affects approximately 1 in 10,000 births [1,2]. The clinical phenotype of the condition has variable severity and age of onset. Patients with SMA are typically categorized into clinical Types 0–IV, where SMA Type 0 is the most severe and Type 1 is the most common among liveborn infants [3]. All subtypes of SMA are caused by DNA variants in the *SMN1* gene, located on chromosome 5q13.2. The majority of the patients (95%) have a homozygous deletion of Exon 7 of *SMN1*, and the other 5% show other DNA variants in *SMN1*, either independently or as a compound heterozygote comprising an Exon 7 deletion on one chromosome and another pathogenic variant in the *SMN1* copy on the other chromosome [2,3]. *SMN2* is a centromeric paralogue that differs from *SMN1* in the Exon 7 region by only five base pairs. The *SMN2* variant, c.840C>T, lies in the coding sequence of Exon 7 and causes skipping of Exon 7 in most *SMN2* transcripts. However, full-length *SMN* protein can be produced from *SMN2*, albeit at very low levels (~10–20%). Thus, *SMN2* can compensate for the *SMN1* deletion to some degree, and at least one *SMN2* gene is retained in all living SMA patients [3,4,5]. The number of *SMN2* copies varies among individuals within the general population, and higher copy numbers are inversely associated with disease in people who do not have a functional copy of *SMN1*; more *SMN2* copies ensures that the absolute amount of *SMN* protein that is produced is higher [6,7,8]. However, genotype–phenotype correlation is not absolute, thus *SMN2* function might not be equivalent in all subjects, indicating that other modifying factors may have a role [9,10].

Upon authorization by the Massachusetts Department of Public Health, the New England Newborn Screening Program of University of Massachusetts Medical School has offered statewide population-based newborn screening for SMA since January of 2018. Being one of the first states to offer screening, we found ourselves facing many unknowns as we worked with colleagues at the U.S. Centers for Disease Control and Prevention (CDC) who were developing assays that states would be able to use upon implementation of high-throughput screening. Specifically, as attractive as the “locked nucleic acid” approach seemed to be for enhancing specificity of primer and probe binding, there was no experience within the high-throughput environment of newborn screening programs at the time and we opted for a more conservative approach. Our screening algorithm uses modifications of two independent CDC-developed RealTime™ PCR-based assays [11,12,13]—and applies them in two assays (Assay A and Assay B) across two tiers [14]. The first tier comprises only Assay A, which is highly sensitive for the detection of absent *SMN1* Exon 7. In the second tier, the tests performed on the eluate from Tier 1 and two new dried blood spot (DBS) punches comprise retests of Assay A and new tests using Assay B. Assay B is designed to determine whether the apparent absence of Exon 7 is due to a hybrid gene [15,16,17,18,19] that carries a variant in the region used to prime the amplification of Exon 7 in Assay A. Results from Tiers 1 and 2 indicating absence of Exon 7 are available the same day and immediately reported with a recommendation for referral for diagnostic evaluation (Figure 1). We also developed a DNA sequencing-based assay to detect the identity of the nucleotide at position c.840 for use in a third tier when Assay A and Assay B results are not fully reconciled. Our SMA sequencing assay is designed to sequence through both *SMN1* and *SMN2* Exon 7, and a region of Intron 6. 

As we first began referring infants for SMA evaluation, it became clear that *SMN2* copy number analysis was not always going to be covered by insurance and, at the time, was taking about two weeks for result turnaround. We sought a solution that did not require another piece of instrumentation. We quickly realized that we could utilize a modification of our sequencing assay to predict *SMN2* copy number in specimens showing absence of *SMN1*. 

We report our findings from the development and implementation of sequencing-based solutions for (1) support of Tier One and Tier Two results and (2) for provision of early and universally available *SMN2* copy number reporting while requisitions for diagnostic copy number testing was being approved and testing performed. Our results reveal the utility of this novel tool for the newborn screening community which provides additional molecular information related to the SMA disorders to aid in the clinical management of the SMA disease in newborn setting. 

## 2. Materials and Methods

### 2.1. Materials

All the materials required to perform sequencing were purchased from Thermo Fisher Scientific (Waltham, MA, USA) including the Big Dye Direct cycle sequencing kit (Cat. 4458688), POP7 polymer (Cat. 4393714), BigDye X Terminator kit (Cat. 4376487). Primers for Exon 7 amplicon of *SMN* encoded genes were purchased from the IDT technology (Coralville, IA, USA). Positive control and carrier DNA for the SMA assays were purchased from Coriell Institute (Camden, NJ, USA). DBS from SMA-affected patients and their parents for development and validation were provided by Biogen (Cambridge, MA, USA). 

### 2.2. DNA Isolation from Dry Blood Spot

DNA was prepared in accordance with our routine protocols as previously published [20].

### 2.3. DNA Primers and Sequencing 

Forward and reverse primers that amplify the *SMN* gene (no discrimination between 1 and 2) are shown in Figure 1B. PCR is performed as follows; 35 cycles at 96 °C for 3 s, 62 °C for 15 s and 68 °C for 30 s, followed by extension at 72 °C for 2 min. PCR products were sized by gel electrophoresis, followed by sequencing according to the manufacturer’s instruction. Variant Reporter™ Software was used for the genetic analysis of Exon 7, using the wild-type *SMN1* gene as the reference sequence.

### 2.4. Validation of SMN1 Sequencing Assay

Thirty reference specimens with various genotypes (affected and parent) provided by Biogen were selected and coded for the validation. Upon completion, decoding was compared with the diagnostic genotype at c.840. A total of 100% concordance was required to pass.

### 2.5. Modification of Sequencing Assay for Determination of SMN2 Copy Number 

DNA from a specimen previously characterized as having one copy of *SMN1* for each single copy of *SMN2* (1:1) was prepared and the DNA concentration was quantified using QUBIT (Life Technology). Likewise, DNA from each of the specimens of SMA-affected patients that had been previously characterized for *SMN2* copy number (reference test specimens; *n* = 7) was also prepared and quantified using QUBIT fluorometer. To determine the copy number of *SMN2* in the test specimen, an equal amount of DNA from the test specimen was first mixed with that of the 1:1 specimen. This mixed DNA was then subjected to PCR using the same PCR conditions described above, followed by bidirectional sequencing. Reference purified DNA purchased from Coriell was used as an accuracy control. Variant Reporter™ Software generated traces used for quality assessment and genotyping. The genotyping of mixed DNA allowed the calculation of the relative ratio of *SMN1*:*SMN2*, and extrapolation from that to define the copy number of *SMN2* genes in the specimen of the SMA-affected patient. Two independent methods, an automated mixed sequence reader program and a manual fluorescence intensity calculation, were used to determine the *SMN2* copy number. The automated mixed sequence reader detects heterogeneity in chromatographic traces and will identify the mixed bases between the homologous sequences (*SMN1*/*SMN2*) and calculates the mixed base ratio within the sequences (http://dx.doi.org/10.1100/2012/365104; 15 July 2021). For the manual calculation, the traces were imported to the Variant Reporter™ Software, and the fluorescence intensity of “C” and “T” nucleotides at the c.840 position of *SMN* gene was recorded along with the background, and the ratio is calculated using mathematical formula A-N/B-N, where A is fluorescence intensity of nucleotide “T”, B is fluorescence intensity of nucleotide “C”, and N is background noise of the sequencing. The final copy number for *SMN2* was achieved by reducing the average calculated copy number by one because one copy of *SMN2* was from the mixing. For example, if the average *SMN2* copy number calculated after analysis is four, the final *SMN2* copy number present in the specimen is three (4-1). 

## 3. Results

Table 1 and Figure 2 show results from the validation of Sanger Sequencing method using the 30 previously characterized specimens. All specimens from the previously characterized SMA-affected patients (*n* = 14) showed nucleotide “T” at c.840 position of *SMN.* As expected, sequencing of specimens from 16 parent specimens showed both “C” and “T” nucleotides at position c.840 (Table 1, Figure 2). Note, in two of the 16 parent specimens, mixed bases at c.840 were clearly noticeable, but the analysis software only noted the presence of a “T” nucleotide. This is due to the software’s cut off set to make a call for mixed bases. We validated a manual correction prior to clinical use. The results from these 30 specimens validated the SMA sequencing assay to differentiate individuals affected with SMA from individuals who possess one or more copies of an intact Exon 7 in the *SMN1* gene. All nine infants who were diagnosed with SMA after an out-of-range NBS result were shown to have only the “T” nucleotide at c.840 of the *SMN* gene (Figure 3). Sequencing results followed the report of an out of range SMA NBS to pediatrician result by a few days.

During the validation of sequencing, we noted that unaffected SMA specimens reported to have different copy numbers of *SMN1* and *SMN2* behave differently at c.840 position of *SMN1* in terms of fluorescence intensity of nucleotide “T” or “C”. We took advantage of this differential fluorescence intensity, and hypothesized that if we mixed the DNA from an individual with a known ratio of *SMN1*:*SMN2* (1:1) with the DNA of an SMA-affected individual *SMN1:SMN2* (0:?), the fluorescence intensity at c.840 or neighboring mixed bases position might provide a ratio of two nucleotides that could be used to extrapolate the *SMN2* copy number in the specimen of the affected individual (Figure 4). 

For validation of our copy number assay, nine well-characterized dried blood spot specimens provided by Biogen and for which we had confirmed genotypes with copy numbers were selected (Appendix A). Seven were reference specimens from SMA-affected patients (reference test specimens) and two were parent specimens with a 1:1 ratio of *SMN1* and *SMN2* (1:1), which were to be used for mixing with the test specimens. DNAs from each of the reference test specimens were mixed with a 1:1 specimen, sequenced, and fluorescent-intensity analysis was performed using two independent methods as described in materials and methods. The observed copy numbers were rounded to the nearest integer and matched exactly with the copy numbers that had been reported for each of the reference patient specimens (Figure 5). Similar results were observed when using another source of 1:1 DNA. (Appendix A). Table 2 shows our *SMN2* copy number findings for the 9 SMA-affected infants who were identified by newborn screening compared to results obtained from diagnostic testing using digital droplet PCR. Results from six of the infants matched, including one who we reported as “at least three copies” and diagnostic testing reported four copies. For the other three infants, our results showed three copies of *SMN2* while the results of diagnostic testing showed only two copies. 

## 4. Discussion

Our sequencing assay has provided invaluable reassurance when the results from our Assays A and B are not fully concordant with no confirmed explanation (e.g., poor extraction, poor amplification, *SMN1* hybrid). We modified our utilization of an SMA sequencing assay for the determination of *SMN2* copy number to provide an early estimate or parallel finding of the copy number that would be obtained by diagnostic testing. We recognize that our copy number assay has some limitations: because we are relying on the DNA from the 1:1 specimen to contribute an *SMN1* copy to determine a relative ratio, its one copy of *SMN2* is also added to the affected SMA specimen and our assay cannot discriminate among copy numbers that are more than four. In addition, our assay is only semi quantitative in nature, yielding a copy number that is always approximate to the absolute ratio (e.g., three copies derived from ratios of 3.25 or 2.75). In light of these limitations, our copy number assay should be an assay of second choice and its use in NBS should be followed by another copy number assay on an independent specimen. 

Specialists are increasingly relying on copy number analyses for prognosis and management of infants who are being identified with SMA by NBS before classical signs and symptoms present [21]. Early on, four or more copies of *SMN2* disqualified a patient from treatment so discrimination among copy numbers higher than four was not immediately critical to the management of the infant and our approximations seemed sufficient for short-term decision making [22]. That said, the variety of assays being used for copy number assessment is wide and we note recent publications documenting discrepancies among laboratories using a single method (MLPA) and different lots of a kit [23,24,25,26]. Our own experience shows that three of nine SMA-affected infants had NBS DBS specimens with *SMN2* copy number results that are discordant from those of diagnostic testing. In still other cases of SMA, discordant genotype–phenotype correlations have been attributed to intragenic *SMN2* variants that influence the amount of full-length SMN protein [27]. Thankfully, we have entered a new era in which SMA-affected infants are identified early and offered new treatments. However, this means that classical characterization of an infant’s innate SMA type by clinical presentation is no longer effective. Without accurate measures and documentation of an infant’s baseline genotype prior to treatment (e.g., how many *SMN2* paralogs are there? How many and which intragenic *SMN2* variants are present?), we will never know the effectiveness of the treatments given and some infants could be undertreated or overtreated. It is incumbent upon screening and diagnostic laboratories to ensure accurate measurements.

In our validation, we compared our results to those well characterized specimens and observed 100% concordance. From the time we began providing copy number results on patients we referred to specialists, we observed some minor discordances between our assay results and those from some external commercial laboratories, which is not surprising. There is a need for well-designed studies of available copy-number assays, which will facilitate the availability of proficiency testing programs for those laboratories seeking to offer copy number testing. Despite its limitations, this novel assay provides reasonable results that can be useful to NBS programs and specialists until confirmatory copy number results are available. We found that recommendations from our results were consistent with those from diagnostic laboratory testing in seven of the nine SMA infants who qualified for treatment by the 2018 guidelines and with nine of the nine who qualified by the 2020 guidelines.

## 5. Conclusions

Since our statewide implementation in January 2018, our use of a sequencing assay and its modification for *SMN2* copy-number analysis has facilitated the early referral and universal access to Appendix A used by specialists in the diagnosis and treatment of infants with SMA. 

## Figures and Tables

**Figure 1 IJNS-07-00047-f001:**
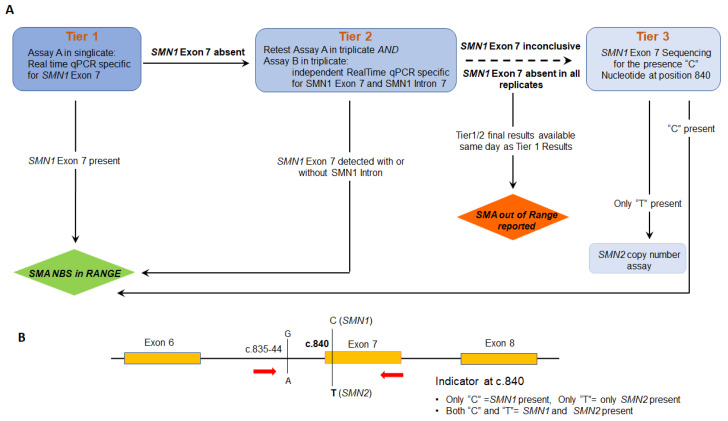
(**A**) Flow chart of current SMA screening algorithm at New England Newborn Screening program. (**B**) Primer locus targeting *SMN1* gene during SMA sequencing. Primers were tagged with M13, and indicated in bold. SMA seq-F- 5′-**TGTAAAACGACGGCCAGT-**AACCTTAACTGCAGCCTAATAATTG-3′ and SMA seq-R- 5′-**CAGGAAACAGCTATGACC-**GCTGGCAGACTTACTCCTTAAT-3′ were used for sequencing.

**Figure 2 IJNS-07-00047-f002:**
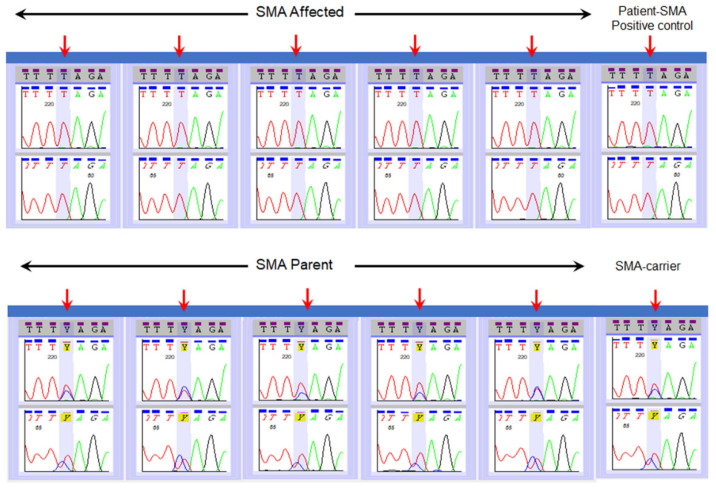
Typical examples of SMA sequencing assay on reference samples from SMA affected patients and their parents.

**Figure 3 IJNS-07-00047-f003:**
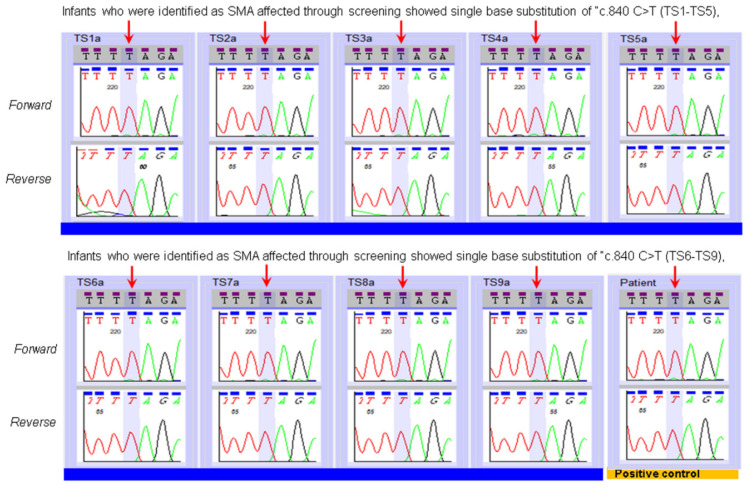
Typical example of SMA sequencing of specimens from nine SMA-affected infants who we referred to specialty centers through screening; all showed single base substitution of c.840C>T.

**Figure 4 IJNS-07-00047-f004:**
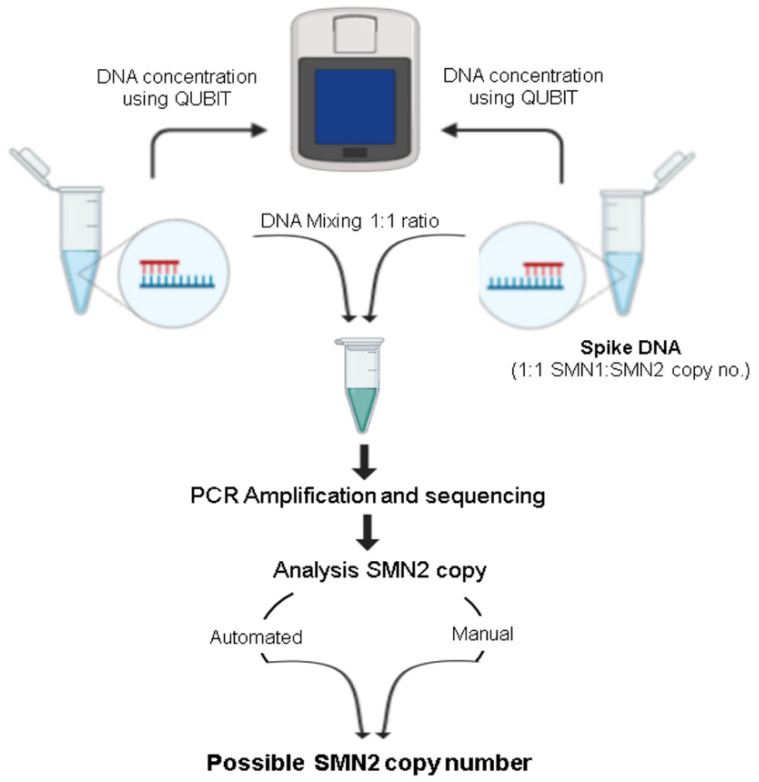
Flow diagram for determination of *SMN2* copy number of SMA affected specimen. DNAs were measured using QUBIT, and equal amount of spike DNA and test DNA were mixed, followed by bidirectional sequencing.

**Figure 5 IJNS-07-00047-f005:**
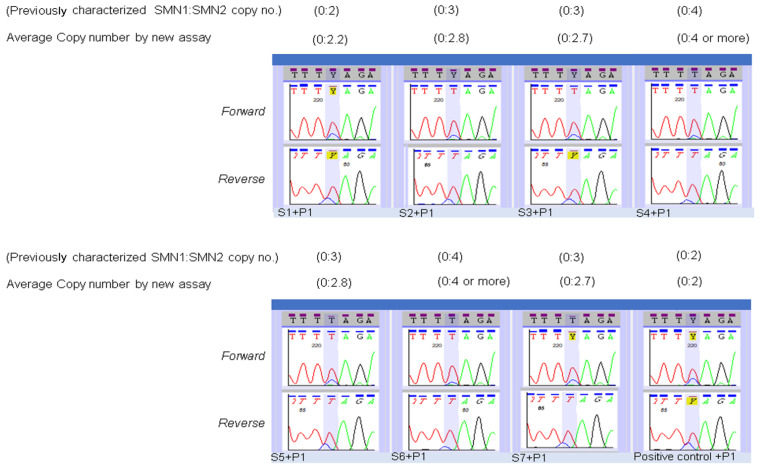
Validation of use of modified sequencing for “determination of *SMN2* copy number” using eight previously characterized SMA affected specimens. DNA from SMA affected specimens, and a Coriell DNA (positive control) were mixed in equal concentration with spike DNA, and bidirectional sequencing was performed. The *SMN1* gene sequence was used as the reference sequence, and data were analyzed to determine the *SMN2* copy number in the test specimen as described in the methods section.

**Table 1 IJNS-07-00047-t001:** NENSP genotype determinations of reference specimens used for validation of SMA sequencing.

Type of Reference Specimens	Number of Specimens	NENSP Detection of Nucleotide at c.840 of *SMN1*
**Affected**	*n* = 14	“T”
**Parents**	*n* = 16	“C” and “T”
**Carrier**		“C” and “T”
**Positive control (known SMA)**		“T”

**Table 2 IJNS-07-00047-t002:** Comparison of *SMN2* copy numbers in newly diagnosed infants generated by our modified sequencing assay and by diagnostic laboratories.

*SMN2* Copy Number Assay Applied to the Specimens of Infants Identified by NBS as SMA Affected.			
MA Patient Specimens	SMA Status	*SMN2* Copy Number Prediction by NENSP from DBS	*SMN2* Copy Number Prediction by Diagnostic Lab Test of an Independent Specimen
**TS1**	Affected	3	2
**TS2**	Affected	More than 3 copies	4
**TS3**	Affected	2	2
**TS4**	Affected	3	2
**TS5**	Affected	2	2
**TS6**	Affected	2	2
**TS7**	Affected	4	4
**TS8**	Affected	3	2
**TS9**	Affected	2	2

## Data Availability

Data beyond what is presented in the article are not publicly available as they are derived from private health information.

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
