# Peer review of "Novel Modification of a Confirmatory SMA Sequencing Assay that Can Be Used to Determine SMN2 Copy Number"

_2409-515X, 2021, doi:10.3390/ijns7030047_

Round 1
Reviewer 1 Report
The article „Novel modification of a confirmatory SMA sequencing assay that can be used to determine SMN2 copy number” by Kumar et al. introduces a modified SMN sequencing protocol that confirms the SMN1 exon 7 deletion and aims at estimating the SMN2 copy number at the same time. The authors present the results obtained for nine infants.
The article was interesting to read, however, I believe that it needs some improvement and discussion. When reading the article, I noticed that obviously insurance coverage is not necessarily comparable in different countries (line 76) so I realized that there might probably be a need for a very simple method in very rare instances. However, I think this should only be a compromise in very rare cases. I therefore think that the authors should definitely point out and discuss that e.g. MLPA is much more reliable as far as SMN2 copy number determination is concerned with very high accuracy if carried out in an experienced lab and well within a timeframe of app. a week. I realize that a new blood sample is required but I do believe that results from NBS should always be confirmed by analysis of a second sample. This is also in accordance with the current recommendations for SMA NBS in Germany (for reference see: Vill et al. Orphanet J Rare Dis (2021) 16: 153). So the described protocol may be of some benefit in a few cases but in general, I think more reliable methods to determine SMN2 copy number should be favoured. I doubt that the protocol is suited for a wider usage as an exact determination of SMN2 copy number is crucial for the choice of therapy. The authors do not compare their results to those obtained with other methods and this should definitely be done.
In addition, I suggest improving and shortening the article. I think it is rather redundant to show the very same sequencing data of patients and carriers nine times each (figures 2 and 3, table 1), it does not really add to the information given. The presentation of the copy number estimation, however, clearly shows the limitations of the method (table 2). As far as figure 1A is concerned, I would ask for some modifications. In case a result prompts tier 2/3, there should be an arrow indicating this step. Furthermore, I think there is some confusion in case SMA is confirmed, looks like a circle to me on the right-hand side where you inevitably have to continue with tier 3 again and again…. Maybe some misunderstanding on my side but not very clear and straightforward….
Make sure you explain every abbreviation: l. 59 CDC
There are quite a few unnecessary blank spaces, e.g. l. 41 SMN1, either
Check for correct punctuation, l. 18
Author Response
Thank-you for the opportunity to submit a revised version of the manuscript entitled “Novel modification of a confirmatory SMA sequencing assay that can be used to determine SMN2 copy number” to the International Journal of Newborn Screening’s Special Issue "Newborn Screening for Spinal Muscular Atrophy".
We appreciate your comments, and have made the requested revisions to the text, tables, and figures, added and replaced references, and made further clarifications. To facilitate your review, our general response in purple, and new text that we inserted into the manuscript in red bold font, highlighting the lines where we inserted the new text with yellow.

Reviewer 2 Report
The manuscript by Kumar and co-workers describes a complementary tier of a previous validated method to apply in a newborn screening program at the University of Massachusetts Medical School that has already offered 56 statewide population-based newborn screening for SMA since January of 2018. The aim of the publication is mainly and merely technical and the third tier proposed by the authors should be validated at large scale to consider methodological pitfalls and biological issues of each patient related to discordance in the genotype-phenotype correlations.
My specific comments are as follow
- General comment. The main concern of this publication to be as an article is the lack of large scale validation of the results of the third tier. SMN2 copy number is important to help in communication and decision making of treatments in positive cases, thus the information cannot be ambiguous or imprecise. For dissemination purposes, the authors may claim the method modification in a short document accounting that the main issues are already published as papers or other documents and have been previously discussed (ref. 11-13)
- I have some difficulties to ascribe the comments and sentences of the Intro with some of the references provided. I recommend being more precise when a reference supports a statement or a concept. Examples: references 9 and 10 are not really linked with the fact that SMN2 is a genetic modifier and the number of copies correlates with the genotype because of the production of more SMN protein. A more general reference such as Feldkötter et al, 2002 Am J Hum Genet or even Calucho et al., 2018 Neuromusc Disord accounting for thousands of cases from the literature can be included.
- SMN2 is a centromeric paralogue that differs from SMN1 by only five base pairs. Actually the concept is that 16 paralogue sequence variants differentiate SMN1 than SMN2. Beyond SMN2 copies, there is a now tendency to highlight the importance of PSVs, rare variants and structure in the consideration of SMN2 by NGS methods (see for example Blasco-Pérez et al., 2021 Hum Mut. There are also other approaches like Ruhno et al., 2019, Hum Genet and Wadmann et al et al., 2020 Brain Communications). This issue may be included in the DIscussion.
- More issues for the Discussion: What is the capability of the method to detect hybrid genes? What is the cost and days of delay when applying the third tier? Please comment that -44 A>G transition may be present in some SMA patients (see ref. Wu et al., Hum Mol genet 2017).
- References on methodologies to determine SMN2 structures, hybrids and copies can be updated (14 to 18)
- It would be interesting to add some Discussion on the limitation and steps to recommend for immediate therapeutic decisions.
- Illustrations (Figure 2 and 4) should be more clear/precise in the way in which SMN2 are semi quantified or modified accordingly.
Author Response
"Please see the attachment"

Reviewer 3 Report
The manuscript written by Kumar et al. described a novel modification of a confirmatory SMA sequencing assay that can be used to determine SMN2 copy number. They had an idea that “C” nucleotide at c.840 of spike DNA from a carrier with one SMN1 copy can be used as a good standard for SMN2 copy number analysis. Their idea is very interesting; their description is straightforward and concise. But I have a single concern on the figures with DNA sequences. I would like to ask them to revise their paper by adding some explanation to the current version.
[Major issues]
The authors do not show the data set of bidirectional sequencing in some samples, which may cause concern for the reviewer.
The authors obtained two DNA sequences using a forward primer and a reverse primer in their experiment, and they converted the DNA sequence obtained by the reverse primer into its reverse-complement counterpart.
In the figures with DNA sequences of this paper, in most cases, upper boxes may show the data obtained using the forward primer, lower boxes the reverse-complement counterparts of the data obtained using the reverse primer.
In the reverse-complement counterparts in the system the authors used, the nucleotides may be denoted by italicized letters (I am sorry if I am wrong). In addition, the shapes of the nucleotide peaks obtained using a reverse primer become different from those obtained using a forward primer. Thus, we can differentiate between the two sequence data obtained by bidirectional sequencing.
However, in Figure 1, the upper and lower boxes of TS 1a show only the data obtained using the forward primer. In Figure 5, the upper and lower boxes of S2+P1 and S7+P1 show only the data obtained using the forward primer, respectively. In Figure S2, the upper and lower boxes of S6+P2 show only the data obtained using the forward primer.
Although they did not say that they will show the data set of bidirectional sequencing in all samples, they showed the data set of bidirectional sequencing in some samples, but not in other samples. Without proper explanation to the figures, readers may doubt that the authors calculated SMN2 copy number using only “good” data obtained in their experiments.
Author Response
" Please see the attachment"

Round 2
Reviewer 1 Report
You have improved the paper and answered all questions to my satisfaction. I realize that not only the applied methods vary significantly worldwide but financing diagnostics is also different and some alternative methods may occasionally be needed. I still feel that the described method should only be second choice and that should be stated clearly in the paper. I strongly suggest participating in quality schemes (such as e.g. EMQN) to ensure the standard of the method. This is also required of labs using for example MLPA and helps to minimize wrong copy number assessment. And just a brief remark: I am convinced that in the hands of an experienced lab, copy number assessment of SMN2 is very reliable. However, to offer alternative methods for selected occasions, publication of the method seems justified.
Author Response
"Please see the attachment"

Reviewer 2 Report
From my side and considering the concerns raised by the other reviewers, I am happy with the changes and explanations made in the manuscript.
Author Response
"Please see the attachment"

Reviewer 3 Report
I am pleased to know that the authors responded to my requests and corrected the figures properly.
Author Response
"Please see the attachment"
